# Review of Mechanisms and Treatment of Cancer-Induced Cardiac Cachexia

**DOI:** 10.3390/cells11061040

**Published:** 2022-03-18

**Authors:** Vignesh Vudatha, Teja Devarakonda, Christopher Liu, Devon C. Freudenberger, Andrea N. Riner, Kelly M. Herremans, Jose G. Trevino

**Affiliations:** 1Department of Surgery, Virginia Commonwealth University School of Medicine, 1200 E Broad St., P.O. Box 980011, Richmond, VA 23219, USA; vignesh.vudatha@vcuhealth.org (V.V.); devarakondstv@vcu.edu (T.D.); liucm3@vcu.edu (C.L.); devon.freudenberger@vcuhealth.org (D.C.F.); 2Department of Surgery, University of Florida College of Medicine, 1600 SW Archer Rd, P.O. Box 100287, Gainesville, FL 32610, USA; andrea.riner@surgery.ufl.edu (A.N.R.); kelly.herremans@surgery.ufl.edu (K.M.H.)

**Keywords:** cancer, cardiac cachexia, reactive oxygen species, TNFα

## Abstract

Cancer cachexia is a multifactorial, paraneoplastic syndrome that impacts roughly half of all cancer patients. It can negatively impact patient quality of life and prognosis by causing physical impairment, reducing chemotherapy tolerance, and precluding them as surgical candidates. While there is substantial research on cancer-induced skeletal muscle cachexia, there are comparatively fewer studies and therapies regarding cardiac cachexia in the setting of malignancy. A literature review was performed using the PubMed database to identify original articles pertaining to cancer-induced cardiac cachexia, including its mechanisms and potential therapeutic modalities. Seventy studies were identified by two independent reviewers based on inclusion and exclusion criteria. While there are multiple studies addressing the pathophysiology of cardiac-induced cancer cachexia, there are no studies evaluating therapeutic options in the clinical setting. Many treatment modalities including nutrition, heart failure medication, cancer drugs, exercise, and gene therapy have been explored in in vitro and mice models with varying degrees of success. While these may be beneficial in cancer patients, further prospective studies specifically focusing on the assessment and treatment of the cardiac component of cachexia are needed.

## 1. Introduction

Cachexia is a multifactorial and multi-organ syndrome characterized by sarcopenia, inflammation, and negative protein balance that cannot be reversed with conventional nutritional support [1,2,3,4,5]. Per international consensus, cachexia is diagnosed if the patient meets one of the following criteria: weight loss > 5% over six months, BMI < 20 and any weight loss > 2%, or appendicular skeletal muscle index indicative of sarcopenia and any weight loss > 2% [2]. Cachexia is noted in many chronic inflammatory conditions, including autoimmune disorders, chronic lung diseases, acquired immunodeficiency syndrome (AIDS), congestive heart failure (CHF), and cancer [3,5]. Cancer cachexia can have a detrimental impact on patient mortality and quality of life. Roughly 50% of cancer patients develop cachexia, with the highest incidence in the gastric and pancreatic cancer patient population [4,5]. In the clinical setting, patients present with lack of appetite, involuntary weight loss, and progressive physical impairment [4,6]. The resulting debilitation can impair the patient’s immune system and reduce tolerance to chemotherapy. Cachexia can also promote hepatic dysfunction, thus perpetuating the nutritional deficit and impacting cardiac and respiratory muscles leading to cardiopulmonary failure [7,8]. Patients that undergo surgical intervention are at increased risk for postoperative complications and have higher mortality rates [9,10]. Overall, cancer cachexia is a predictor of poor patient outcomes and is responsible for roughly 20% of cancer-associated deaths [4].

There have been many studies investigating the molecular pathogenesis of cancer-associated cachexia, particularly in skeletal muscle. The main driving force behind this syndrome is chronic systemic inflammation and tumor–host interaction. Tumor and host immune cells release pro-cachectic cytokines such as tumor necrosis factor (TNFα), interleukin 1 (IL-1), IL-6, and interferon gamma (IFNγ) [1,6,7,11]. TNFα drives skeletal muscle catabolism by inducing ubiquitin-mediated proteasome degradation (UPR) via the nuclear factor kappa-B (NF-κB) pathway [7,11,12]. TNFα also synergizes with IL-1 and IFNγ to impact appetite. They cross the blood–brain barrier and induce a series of neurohormonal alterations to promote anorexia and muscle wasting. They increase levels of available serotonin, reduce secretion of appetite-stimulating hormones such as neuropeptide Y and ghrelin, and trigger the hypothalamic–pituitary–adrenal axis, thus further promoting skeletal muscle and adipose tissue breakdown [7,8,11].

While there is heavy emphasis on skeletal muscle degradation, cancer cachexia can also have an impact on cardiac muscle and function. Independent of cardiotoxicity secondary to chemotherapy, cancer cachexia can cause cardiac muscle degradation, leading to heart failure [13,14]. Understanding the mechanisms behind cancer-induced cardiac cachexia can allow for improvements in patient management both in the setting of chemotherapy and surgical intervention. This article will review the current literature regarding cancer-induced cardiac cachexia, including mechanisms, diagnostic methods, and current therapeutic options.

## 2. Materials and Methods

A literature search was performed to identify original research pertaining to cardiac alterations secondary to cancer cachexia. An advanced search was performed on the PubMed Search Engine from 1960 through to 24 August 2021. The following search string was utilized: ((cardiac cachexia) AND (cancer)) OR ((cancer cachexia) AND (heart)) OR ((cardiac sarcopenia) AND (cancer)) OR ((cardiac cachexia) AND (methods)) OR ((cardiac sarcopenia) AND (methods)) OR ((cardiac muscle wasting) AND (cancer)) OR ((cardiac muscle wasting) AND (methods). Studies were limited to journal articles and those published in English. Studies were excluded if they were abstract only, not published in full, or were duplicate articles. Articles were considered eligible if the research addressed cardiac cachexia in the setting of cancer. Studies that pertained to cardiac toxicity secondary to chemotherapy or heart failure precipitating cachexia were excluded.

Two individuals (T.D. and V.V.) independently screened titles and abstracts based on the above search terms and inclusion/exclusion criteria. Articles were initially screened for relevance by title. The remaining articles were screened by abstract for eligibility. All articles that met inclusion criteria were independently reviewed by both reviewers in their entirety.

## 3. Results

The resulting search yielded 1383 articles including duplicates (Figure 1). In total, 121 articles were selected based on title relevance. These articles were then screened based on either abstract or full text. Ultimately, 70 articles were identified that presented original research on cancer-induced cardiac cachexia.

## 4. Discussion

### 4.1. Mechanisms

#### 4.1.1. TNFα

Pro-inflammatory cytokines are known to play a critical role in the development of cancer cachexia. Markers such as TNFα, IL-6, and Ataxin-10 were observed to be elevated in cardiac tissue of colon-26 (C26) adenocarcinoma mice [15,16,17]. In early studies, TNFα was noted to have an impact on protein degradation rates in cardiac muscle after observing a decrease in protein loss in hepatoma mice receiving anti-TNF treatment [18]. Similarly, increased myocardial weight loss was observed in normal female rats treated with TNFα [19]. TNFα activates the NF-κB pathway, resulting in increased myocardial turnover via the UPR system and increased oxidative stress secondary to suppression of glutathione peroxidase production [20]. Furthermore, other moieties, such as high mobility group box protein 1 (HMGB1), which induce TNFα expression in cardiac myocytes were also increased in cancer cachexia models [21]. As in skeletal muscle, TNFα plays a pivotal role in cardiac cachexia via multiple pathways. 

#### 4.1.2. Autophagy

Increased autophagy is another mechanism involved in cancer cachexia. Autophagy involves degradation of intracellular proteins, organelles, and macromolecules secondary to cellular stress. In colorectal tumor-bearing mice, markers of increased lysosomal activation, including cathepsin L, beclin, and microtubule-associated protein light chain 3 (LC3), were elevated in the myocardium. Direct observation of autophagy was obtained via electron microscopy, demonstrating autophagic vacuoles containing mitochondria and cytoplasm [22]. Autophagy can be activated by multiple pathways, including alterations in AMP-activated protein kinase (AMPK) and PI3K/Akt/mTOR signaling.

Activation of phosphoinositide-3 kinase (PI3K) by insulin and insulin-like growth factor 1 (IGF-1) leads to downstream upregulation of Akt and mTOR, which subsequently enhances protein synthesis. In cancer patients, the utilization of glucose by tumor cells leads to reduced circulating insulin, thus reducing activation of this pathway. As confirmation, mTOR expression was shown to be reduced in preclinical models of colorectal cancer (CRC) and melanoma [20,23]. One of these studies, however, noted that while mTOR expression was suppressed, there was an increase in Akt phosphorylation. This pathway activation could be a compensatory mechanism by the myocardium to preserve cardiac mass [23].

AMPK is typically activated by decreased intracellular ATP/ADP ratio and serves to suppress energy demanding cellular activities. AMPK activation can induce upregulation of autophagy associated moieties and suppress mTOR formation, thus reducing protein synthesis. Manne et al. demonstrated the upregulation of AMPK and decrease in mTOR signaling in the cardiomyocytes of adenomatous polyposis coli mice (APC+). Interestingly, they also noted no evidence of increased protein ubiquitination or apoptosis in these hearts, thus placing greater weight on autophagy as a key molecular mechanism in cancer-induced cardiac cachexia [23]. 

#### 4.1.3. Reactive Oxygen Species

Increased oxidative stress is the result of an imbalance between the production of reactive oxygen species (ROS) and their breakdown by antioxidant mechanisms, such as glutathione peroxidase. Elevated ROS can promote activation of proteolytic pathways such as ubiquitination and calcium-dependent proteolysis [24]. Hinch et al. demonstrated this imbalance in pro- and antioxidants in murine adenocarcinoma cell lines 13 (MAC13) mice by noting increased levels of xanthine oxidase and decreased expression of antioxidant superoxide dismutase in the myocardium [25]. This finding is corroborated by Lee et al., whose study showed increased ratios of oxidized to reduced molecules and decreased levels of ROS scavengers in Lewis lung carcinoma bearing mice. The resulting ROS also damages mitochondrial DNA, which can negatively impact the myocytes’ ability to process oxygen in the electron transport chain, thus accumulating more ROS [26]. 

#### 4.1.4. Ubiquitination: Atrogin-1/MuRF1

Protein degradation in both skeletal and cardiac muscle requires ubiquitination of targeted proteins. These are typically done via E3 ligase enzymes. Atrogin-1 and MuRF1 (muscle ring finger 1) are two ligases that have been implicated in cardiac muscle proteolysis. These enzymes can be attenuated by many pathways, including NF-κB and Akt/mTOR. In their studies, both Tian et al. and Wysong et al. demonstrated increased expression of both ligases in the atrophic myocardium of C26 mice [27,28]. Matsuyama et al. also analyzed cytokine and Atrogin/MuRF expression levels in C26 mice. However, they also compared the impact of both intraperitoneal and subcutaneous implantation. Interestingly, the subcutaneous group had no reduction in cardiac weight and normal levels of both ligases in contrast to the intraperitoneal group. These findings demonstrate that tumor location plays a significant role in cancer cachexia phenotype and progression [29].

#### 4.1.5. Cancer-Induced Cardiac Cachexia Pathways

There are many other mechanisms in the literature that have been implicated in cancer induced cardiac cachexia, including calcium-dependent proteolysis, decreased neural stimulation, activin activation, and metalloproteinases. Costelli et al. demonstrated the role of Ca^2+^-dependent proteolytic systems in cardiac atrophy. In this pathway, increased intracellular calcium leads to elevated levels of proteolytic enzymes, such as calpain. In their study, they evaluated cachexia in male Wistar rats inoculated with ascites hepatoma (AH-130) cells. Analysis of the cardiac muscle revealed elevated levels of calpain and decreased expression of Ca^2+^-ATPase (transport protein) and calpastatin (calpain inhibitor). However, the authors also note that calpains have been implicated in degradation of transcription factors such as NF-κB, thus suggesting that calpains may play a more complex role in attenuating myocardial damage [30,31]. Muhlfeld et al. noted decreased levels of nerve growth factor in stellate ganglion of myocardium derived from Lewis lung carcinoma bearing mice. This was associated with a decrease in vesicle number per axon and total length of axon in the left ventricle (LV). While the study noted the potential role of elevated IL-6 and TNFα, no causal relationship was established [16]. Activin molecules, which are TGF-B moieties, have been implicated in both cardiac and skeletal muscle wasting. These molecules bind to activin receptor type 2B (ACVR2B) and can induce degradation of skeletal muscle, cancellous bone, and cardiac function. Antagonism of ACVR2B has been shown to preserve cardiac function in CRC-bearing mice [32]. This pathway is further modulated by histone deacetylases, such as SIRT6, that block the expression of activin and ACVR2B [33]. Finally, one other pathway that has not been studied extensively is the alteration of the extracellular matrix (ECM). Devine et al. demonstrated that the cardiac and skeletal muscles in C26 bearing mice had increased protein levels of matrix metalloproteinases. These enzymes degrade the ECM, causing disruptions in cell-to-cell and cell-to-basement membrane interactions, thus promoting cardiac dysfunction. While this mechanism has been discussed in the setting of heart failure, this is the first study to address its role in cancer-induced cardiac dysfunction [34].

### 4.2. Models to Study Cardiac Cachexia

Establishing a reproducible animal model of cachexia is crucial to delineate the mechanisms leading to cardiac sarcopenia. Historically, the Lewis lung carcinoma, C26 colorectal adenocarcinoma, patient-derived xenograft, and genetically engineered mouse models have been used to investigate cachexia [35]. To standardize the well-studied C26 model for studying cancer cachexia, Bonetto et al. subcutaneously injected 1 × 10^6^ C26 cells in each CD2F1 mouse. At the end of a 10–14-day period, significant losses in skeletal and cardiac muscle weights were observed [36]. In a subcutaneous Ehrlich ascites carcinoma (EAC) model in 129/SvJ mice, echocardiography performed at weekly intervals showed a decline in left ventricular (LV) thickness, ejection fraction (EF) and Fractional Shortening (FS) and an increase in LV internal diameter [37]. A pancreatic ductal adenocarcinoma (PDAC) cachexia model was developed by generating a tumor cell line from genetically modified mice with oncogenic KRAS G12D mutation and additional tumor suppressor P53 R172H mutation, and subsequently implanting the cells in wild-type C57BL/6 mice either subcutaneously, intraperitoneally or orthotopically. This study demonstrated PDAC induced cardiac cachexia in an autophagy-dependent manner. Additionally, mice with orthotopic and intraperitoneal implants demonstrated a reduction in cumulative food intake when compared to mice receiving subcutaneous implants [35].

Orthotopic tumor mouse models can also involve implantation of patient-derived xenografts into tissues matching the tumor histology, thereby preserving the tumor architecture and stromal environment. However, the requirement of immunosuppression and the potential loss in signaling pathways due to absence of species homology can potentially serve as limitations toward implementing this model extensively [35].

The lack of a standard approach toward developing a cachexia model poses a significant barrier in comparing existing data surveying this phenomenon. Bonetto et al. noted that the strain of the mouse, method and site of tumor implantation, tumor source and number of cells injected can influence outcomes pertaining to cachexia even within the C26 model [36]. In the EAC model—as in other animal models—cardiac dysfunction manifested as LV dilatation, but cancer cachexia generally presents as reduced ventricular volume in humans, even in the absence of chamber dilatation [37]. Therefore, further standardization and refinement of animal models are needed to achieve translationally viable outcomes. 

### 4.3. Methods to Assess Cardiac Cachexia

Cardiac cachexia in cancer manifests with biochemical and functional impairments, which can be analyzed to monitor disease progression and efficacy of ongoing interventions. In preclinical studies, measurements of cardiac weight at necropsy—generally expressed as a ratio to total body weight—have been utilized to demonstrate cardiac sarcopenia. However, the presence of generalized edema in tumor bearing mice was shown to influence this ratio, reflecting a higher than expected value for normalized cardiac weights in a C26 model at day 14 after tumor implantation [15]. Therefore, metrics utilizing weight-loss relative to BMI as a parameter for cachexia must factor in variability stemming from study design prior to analysis and interpretation.

Imaging techniques, including echocardiography and cardiac MRI, allow for assessment of changes in cardiac functional status in-vivo and in clinical studies. In a retrospective clinical analysis, a positive correlation was observed between BMI and LV and RV wall thickness (LVWT and RVWT) in GI cancer and LVWT in lung cancer [38]. In a C26 model implanted in CD2F1 female mice, changes in fractional shortening and posterior wall thickness (PWT) were noted in tumor bearing mice, but diastolic parameters were not affected [39]. In a comparative clinical study involving patients with colorectal cancer (CRC), CHF and controls, impairment in left ventricular ejection fraction (LVEF) was observed in CRC patients, although not to the degree observed in patients with CHF. Increased PWT was observed in CRC and CHF patients, but not in controls [40]. Alternatively, cardiac MRI might be more sensitive to changes in cardiac structure than can be detected via 2D echocardiography and is operator independent. While MRI studies in cardiac cachexia from cancer are lacking, analysis of CHF patients with and without cachexia showed down-trending cardiac weights in patients with cachexia versus an uptrend in patients without cachexia despite the CHF [41]. Further standardization of data from imaging studies can be attained by tracking indices combining data from individual parameters. For instance, the cachexia index (CXI) was formulated by incorporating skeletal muscle area and skeletal muscle index at the L3 level—as obtained from abdominal CT scan images [42]. However, due to the paucity of imaging data assessing cardiac cachexia in patient populations, further analysis is required prior to development of standardized indices for cancer associated cardiac cachexia. 

Molecular and microscopic techniques offer a unique window into the subcellular changes associated with cachexia. In Tian et al., transverse electron microscopy (TEM) showed disruptions in myocardial ultrastructure [15]. In a C26 model, proteomic assessment of cardiac, soleus and gastrocnemius tissues showed dissolution of Z disc and M line proteins, along with downregulation of proteins involved in myocyte energetics and substrate metabolism [43]. Metabolomic data from a C26 model demonstrated a unique ‘footprint’ to cachexia-associated changes in skeletal muscle that is distinct from the changes seen purely from caloric restriction [44]. Finally, an elevated neutrophil-to-lymphocyte ratio (NLR)—as derived from a complete blood count with differentials—was associated with greater weight loss and cachexia in patients with advanced colon, lung and prostate cancers [45]. The baseline NLR status was also demonstrated to be a negative prognostic biomarker for patients with cachexia in a multicenter cohort study evaluating 2612 patients with cancer [46]. However, the significance of this ratio for tracking cancer associated cardiac cachexia is yet to be explored. Identifying structural and biochemical changes prior to overt clinical manifestation of cardiac cachexia is significant. In Xu et al., while diastolic dysfunction was not seen in echocardiography data, cardiomyocyte function assessed at a cellular level showed impaired relaxation kinetics. While cardiac weights were not reduced, the expression of MAFbx and Bnip3—known biomarkers for degradation in skeletal muscle—were upregulated [39]. Within echocardiography parameters, a decline in global longitudinal strain (GLS) can predate functional decline in EF, as changes in GLS were shown to correlate with a decline in LV mass in a clinical study involving patients with non-small cell lung cancer (NSCLC) [47]. Cramer et al. noted that heart rate variability (HRV), a parameter associated with intact autonomic regulation of the heart, was impaired in patients with CRC. While the CRC patients were not tachycardic, their heart rates were significantly elevated compared to patients with CHF and controls, and the authors speculated a potential utility of monitoring heart rate to assess progression toward clinically symptomatic cardiac cachexia [40]. Taken together, these findings underscore the importance of a multimodal approach in estimating cardiac cachexia in preclinical studies and patient populations. 

### 4.4. Treatment

Potential therapies are wide-ranging and target various pathways known to play significant roles in cancer cachexia. Table 1 briefly summarizes the treatment modalities discussed below.

#### 4.4.1. Anti-Proteolytics

Dysfunctional regulation of proteolytic activity is a target of interest for pharmacologic intervention. Devine et al. found that minocycline, a matrix metalloproteinase (MMP) inhibitor, improved FS and EF in a murine model. Decreased collagen RNA expression confirmed attenuation of MMP mediated cardiac fibrosis [75]. Cathepsin D, a lysosomal protease, is another proteolytic enzyme elevated in tumor-bearing mice. While its inhibition in murine cardiac tissue was shown to be feasible with the aspartyl protease inhibitor Pepstatin in a study by Greenbaum and Sutherland, effects on cardiac function have yet to be described [54]. Nevertheless, in vitro studies in primary neonatal rat cardiomyocytes showed inhibition of oxidative stress and resultant apoptosis by Pepstatin A. [55]. Saitoh et al. led a study investigating applications of Erythropoietin (EPO), a hormone used for amelioration of cancer-induced anemia, in cancer cachexia. In their rat model of liver cancer cachexia, EPO-associated increases in cardiac weight, stroke volume (SV), FS, and physical activity were hypothesized to be attributed to observed decreased levels of Trypsin. High-dose EPO treated rats conferred a survival advantage [65]. While promising, further study is needed to take anti-proteolytic therapy to the clinical stage.

#### 4.4.2. Statins

Applying statins’ anti-inflammatory effects to cancer cachexia has yielded dissimilar results. Muscaritoli et al. surprisingly found that simvastatin had negative effects on muscle wasting in Yoshida AH-13 ascites hepatoma rats and highlighted a 15% reduction in cardiac weight, prompting them to caution its administration in cachectic patients [56]. This was contrasted in a more recent study conducted by Palus et al. in AH-130 bearing rats. They found that simvastatin treatment reduced muscle and body weight loss, improved LVEF, and increased SV. Importantly, treatment reduced mortality. These contradictory findings necessitate additional study to clarify simvastatin’s effects [57]. 

#### 4.4.3. STAT3 Inhibition

STAT3 upregulation has various pro-cancer effects. Inhibition with Cryptotanshinone, a chemical with reported antiproliferative, anti-inflammatory, and anti-tumor properties, was explored by Chen et al. in CT26 tumor-bearing mice. While cellular analysis confirmed Cryptotanshinone induced STAT3 inhibition, treatment also decreased myocardial mass loss, body weight loss, muscle wasting, and epididymal fat [74]. Though understanding is limited, STAT3 inhibition is another pathway target in cancer cachexia. 

#### 4.4.4. Heart Failure Medication

Applying heart failure treatment to cardiac cachexia may alleviate shared symptoms.

To this end, Springer et al. explored bisoprolol (beta-blocker), imidapril (angiotensin converting enzyme inhibitor), and spironolactone (aldosterone inhibitor) in a rat hepatoma model. Treatment with spironolactone and bisoprolol preserved left ventricle mass, body weight, fat mass, and lean body mass in addition to improving LVEF and left ventricular FS with spironolactone showing greater effects. These agents also reduced caspase-3 and ubiquitin proteasome activity. Imidapril failed to produce similar outcomes. Specifically, in spironolactone-treated rats, cardiac fibrosis was reduced, which, when paired with observations of elevated aldosterone levels in human cancer patients, cements aldosterone’s significance in cancer cachexia [76]. These results parallel a related study of spironolactone’s application in an AH-130 hepatoma rat model, which found that treatment preserved LV diameter, increased LV mass, and downregulated neutrophil gelatinase-associated lipocalin, an aldosterone regulated gene upregulated in the heart failure and cachectic environment [77]. 

Angiotensin I and II inhibitors were also explored in cancer cachexia. Stevens et al. investigated losartan, an angiotensin II receptor blocker, in a C26 mouse model and found that losartan preserved EF, SV, and PWT while reducing left ventricular end-diastolic dimensions and normalizing calcium signaling dysfunction observed in tumor-bearing mice. Time to 90% muscle peak-shortening and 90% muscle re-lengthening were shorter in untreated tumor-bearing mice compared to treated mice. Interestingly, losartan impeded tumor cell proliferation [58]. Similar results were observed in an investigation of the steroidal lactone Withaferin A. Used for its anti-inflammatory characteristics and ability to impede tumor growth, Withaferin A administration to cachectic mice with ovarian cancer reduced tumor-associated increases in angiotensin II and subsequently reduced cardiomyocyte cross-sectional area loss, systolic and diastolic dysfunction, and fibrotic deposition [59]. These studies and the overlapping pathophysiology between cardiac cachexia and heart failure merit exploration of other heart failure agents.

#### 4.4.5. Reactive Oxygen Species

Increased levels of ROS are key in cancer cachexia advancement. Smuder et al. recently explored ROS attenuation through restoration of mitochondrial function via the antioxidant peptide SS-31 in C26 mice. They reported that SS-31 decreased ROS production, restored left ventricular function, and diminished proteolytic Calpain activity in the heart when compared to saline-treated mice [52]. Other studies [78] emphasizing mitochondrial dysfunction in skeletal muscle atrophy supports further targeting of this pathway. Also involved in ROS modulation is Ubiquinol, an antioxidant deficient in various types of cancer. While its administration was associated with increased muscle mass in a C26 murine model, treatment failed to improve protein degradation, left ventricular diastolic diameter, PWT, and FS [53]. While in its early stages, development of therapeutics addressing ROS pathways underlying cancer cachexia represents a promising field of study. 

#### 4.4.6. NF-κB

NF-κB is a proinflammatory transcription factor whose activation by tumor derived-cytokines potentiates proteolysis. Shadfer et al. utilized the phytoalexin Resveratrol in a C26 mouse model to modulate NF-κB. In this study, they found that tumor-attributed decreases in cardiac weight to body weight ratios, anterior wall thickness (AWT), and PWT disappeared with Resveratrol treatment. Increases in NF-κB activity in the hearts of tumor-bearing mice were also ameliorated along with MuRF1 mRNA whose role in cardiac atrophy has been well documented [50,51]. Wysong et al. also targeted NF-κB through novel drugs Compound A and NF-κB essential modulator (NEMO) binding domain peptide in C26 mouse models. Separately, Compound A and NEMO binding domain peptide inhibited IkB kinase and subsequently NF-κB in the heart, preserving cardiac mass, EF, FS, cardiomyocyte area, and heart wall thickness [27]. Luteolin, a natural flavonoid, represents another NF-κB inhibiting agent, and its effects were discovered to include preservation of cardiac muscle mass, reduction of TNF-α and IL6 levels, and attenuation of increased MuRF1 levels in a Lewis lung cancer mouse model [48]. Distinctly, Aquila et al. used chemotherapy agents, trabectedin and lurbinectedin, to inhibit cytokine activation of NF-κB in C26 mice. While preliminary analysis of trabectedin and lurbinectedin demonstrated survival benefits, further study into lurbinectedin failed to impact NF-κB signaling in atrophying myotubes or exert cardiac protective effects [49]. Much like other disease processes, normalization of NF-κB activity is a key factor in addressing cachexia.

#### 4.4.7. ActRIIB

Upregulation of the ACVR2B pathway is involved in numerous types of cancer and is associated with cancer cachexia. Zhou et al. found that ACVR2B antagonism reversed muscle wasting, protected against cardiac atrophy, and imparted a survival benefit. Cellular investigation connected the inhibition of ubiquitin-proteasome processes and ubiquitin ligases in muscle tissue with ACVR2B inhibition, thus further defining a mechanism for its effects [60]. The strength of these findings corroborates the ACVR2B pathway’s emergence as a therapeutic target in cancer cachexia. 

#### 4.4.8. Oxypurinol 

Hyperuricemia is a risk factor for worse outcomes in cardiac cachexia. Uric acid regulation through inhibition of xanthine oxidase follows as a logical step to reduce hyperuricemia-related consequences. Oxypurinol (xanthine oxidase inhibitor) increased LVEF, FS, total cardiac weight, and cardiac output in a rat model of cancer cachexia [66]. Though promising, the mechanisms of these effects must be demonstrated in the pre-clinical setting before clinical application. 

#### 4.4.9. Antidepressants

Depression and anxiety are associated with cancer cachexia. Addressing these symptoms may not only provide direct relief but may also have secondary benefits. Elkina et al. explored the utilization of the antidepressant and anxiolytic drug tandospirone in a Yoshida hepatoma rat model. Treatment preserved muscle mass, locomotor activity, and food intake with accompanying improvement in left ventricular mass, SV, EF, and FS. Not to mention, a survival advantage was seen with treatment [67]. This study suggests antidepressants may have benefits outside their intended psychiatric impacts that require further investigation in future studies. 

#### 4.4.10. Weight and Muscle Gain

Promotion of muscle and weight gain have shown promise as therapeutic strategies. Rosiglitazone, an agent for type 2 diabetes mellitus, was found to not only improve LVEF, FS, and cardiac output in a cachectic rat model but also decrease muscle wasting and produce a survival benefit. Of note, previous observations of Rosiglitazone-induced cardiac hypertrophy were not replicated [61]. The application of other therapeutics used in diabetes such as Metformin have demonstrated potential utility in cancer-induced skeletal muscle cachexia mouse models but its significance in cancer-induced cardiac cachexia has never been explored [79,80]. Appetite-stimulating therapeutics like Megestrol acetate have also been investigated to promote weight gain. In cachectic tumor-bearing rats, improved LVEF, FS, and left ventricular end-systolic volume were associated with Megestrol acetate. Above all, mortality benefits were associated with treatment. Decreased Beclin-1 and LC3 indicated that Megestrol acetate suppressed autophagic pathways in cardiac tissue [81]. Formoterol, a potent B2 selective agonist known to generate skeletal muscle growth, was explored in a rat cachexia model. While treatment was associated with a non-significant increase in cardiac weight, preservation of left ventricular diameter and improved end-diastolic and end systolic volumes were all significantly associated with Formoterol [82]. Ojima et al. also studied the effects of promoting muscle growth in cachectic in vivo models by inhibiting negative regulation of skeletal muscle mass growth mediated by growth differentiation factor 8 (GDF-8). They found that by administering peptide 2 in tumor implanted C57BL/6 mice, they could inhibit GDF-8 signaling and promote myoblast differentiation. Although improvements in gross muscle mass with concomitant increase in grip strength were observed, cardiac muscle was excluded from these benefits [83]. These preclinical studies indicate that promoting muscle and weight gain hold potential for therapeutic applications in cancer cachexia. However, the limited data regarding these treatments necessitate further study. 

#### 4.4.11. Testosterone

Adjunct testosterone therapies have positive effects in cancer patients, but their impacts on cardiac function were only recently clarified. In 2019, Scott et al. published an analysis of a previous study assessing testosterone utilization in cachectic squamous cell carcinoma patients and evaluated testosterone’s effects on cardiac atrophy and dysfunction. Their investigation demonstrated testosterone-associated increases in SV and LVEF, as well as improvements in vascular parameters of arterial elastance and ventricular arterial coupling [84]. While promising, high powered randomized trials are necessary to elucidate testosterone’s mechanism of action and confirm its effects in cancer cachexia before clinical application. 

#### 4.4.12. Nutrition

Previous studies have suggested nutritional intervention as an adjunct therapy in cardiac cachexia. One rat model linked leucine’s anti-inflammatory, anti-oxidative stress, anti-proteolytic, and anti-apoptotic properties with attenuation of cancer cachexia-associated cardiac injury. Specifically, leucine decreased chymotrypsin, myeloperoxidase, tissue inhibitor of metalloproteinase, total plasminogen activator inhibitor 1, and caspases 3 and 7; all observed to be elevated in cachectic rats. Decreased levels of Tissue inhibitor of metalloproteinase 1 (TIMP-1), a marker of pathologic myocardial processes, further supported leucine administration [62]. A more recent study targeted oxidative stress mechanisms of myocardial damage stemming from mitochondrial dysfunction with co-administration of lauric acid and glucose. Nugaka et al. showed that this combination reduced mitochondria dysfunction in an in vitro cachexia model, leading to amelioration of oxidative stress and restoration of ATP. Their mice model also demonstrated that lauric acid and glucose improved levels of myocardial atrophy and muscle maturity [63]. Chance et al. separately studied the effects of dietary intervention in rats through maintenance with total parenteral nutrition and co-administration of acivicin (glutamine analog) and clenbuterol (beta-2 antagonist). They found that just sustaining sarcoma rats on TPN led to increased cardiac mass. Adding acivicin and clenbuterol enhanced the aforementioned effect while increasing skeletal muscle mass, inhibiting tumor growth, and promoting protein content [64]. Addition of over-the-counter nutritional supplements and medications may serve as valuable adjuncts to primary therapies. Some mouse model studies have suggested that non-steroidal anti-inflammatory drugs, glutamine, glycine, fish oils, carnitine and creatine may each hold value in the treatment of cancer-induced skeletal muscle cachexia [85,86,87,88,89,90]. However, these therapeutic effects have yet to be replicated or studied in the context of cancer-induced cardiac cachexia. Altogether, these positive results spanning various nutritional strategies merit further exploration. 

#### 4.4.13. Exercise

The benefits of exercise and conditioning in cancer patients are well established and may play a role in mitigating cardiac cachexia [68]. Parry and Hayward investigated the effects of exercise in a rat tumor model developed through inoculation of 13,762 MatBIII breast adenocarcinoma cells. They found that rats with running wheel training had reduced cardiac autophagic activity compared to sedentary tumor-bearing mice. Additionally, treatment impeded tumor growth and attenuated shifts in myosin heavy chain isoforms linked to reductions in left ventricular developed pressure [69]. A similar study in a C26 mouse model of cancer cachexia showed that resistance and aerobic exercise led to improved muscle mass, strength, and mitochondrial function [70]. Fernandes et al. alternatively approached exercise intervention by administering structured aerobic exercise training (AET) in a C26 mouse model. Treated mice had partially rescued cardiomyocyte cross-sectional diameter, improved EF, and a decrease in cardiac remodeling apparent in reduced necrosis, inflammation, and collagen deposition. Furthermore, AET attenuated the upregulation of BNIP3, a gene implicated in cardiac autophagy [71]. Another interventional AET study in a mammary tumorigenesis model advanced understanding of exercise’s mechanisms by identifying decreased cardiac TNF-related weak inducer of apoptosis (TWEAK) and NF-κB signaling, TRAF6, and atrogin-1 as causes for modulation of pathogenic cardiac remodeling [72]. Altogether, these preclinical studies have furthered the case for exercise in cardiac cachexia, and its ease of implementation supports inclusion into standard practice.

#### 4.4.14. Gene Therapy

Gene therapy is a potential therapeutic avenue through targeting aberrant pathways. Winbanks et al. focused on the overactivation of the ActRIIB receptor, known to induce muscle wasting through binding of procachectic factors and downstream SMAD2/3 phosphorylation. SMAD2/3 proteins accumulate during muscle immobilization and are involved in skeletal muscle atrophy. By upregulating SMAD7, a SMAD2/3 negative regulator, with a recombinant viral vector, cachectic mice were protected from skeletal and cardiac muscle wasting via downregulation of ubiquitin ligases associated with SMAD2/3 linked atrophy [73]. Though still emerging, gene therapy has the potential to circumvent the shortcomings of other conventional approaches. 

Research in the treatment of cardiac cachexia is still evolving but has yielded promising therapeutic targets. Continued collaboration between researchers and clinicians is needed to translate the results of current and future animal studies to the clinical setting. 

## 5. Conclusions

Cancer cachexia is a debilitating syndrome that can impact a cancer patient’s quality of life, treatment tolerability, and mortality. The pro-inflammatory state that is characteristic of this condition can negatively impact the cardiac muscle architecture, thus promoting cardiac dysfunction. Despite the research discussed above, there are still many gaps in our understanding of cancer-induced cardiac cachexia, and increased knowledge of its molecular mechanisms can help identify more therapeutic targets. Furthermore, while there are many promising treatment modalities, including pharmacologic, nutritional, and genetic interventions, these are all in the preclinical phase. Further investigations need to be performed to determine the efficacy of these therapies and their long-term safety profiles in the patient population. Successful treatment of this disease process can potentially improve patient quality of life and improve survival. 

## Figures and Tables

**Figure 1 cells-11-01040-f001:**
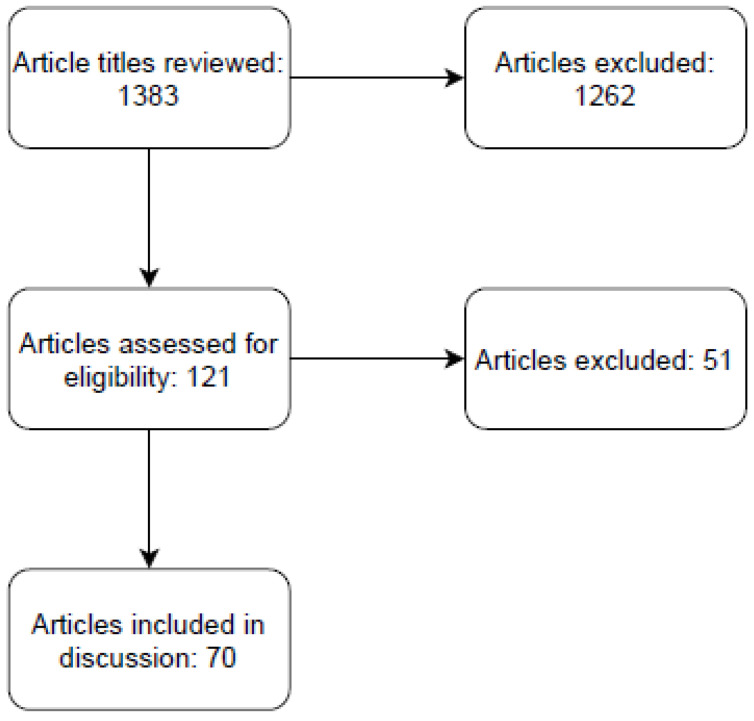
PRISMA diagram of literature search.

**Table 1 cells-11-01040-t001:** Treatments modalities for mitigating cancer-induced cardiac cachexia.

	Treatment	Mechanism	Benefit	Reference
NF-κB Pathway Inhibitors	Compound A, NF-κB essential modulator, Luteolin	NF-κB inhibiting agents	Preserved cardiac mass and EF and reduction of inflammatory markers	[27,48]
	Trabectedin, Lurbinectedin	Inhibits cytokine activation of NF-κB	Some survival benefit but no clear impact on NF-κB signaling in cardiac tissue	[49]
	Resveratrol	Inhibition of inflammatory pathways and improvement in myocardial calcium handling	Reduction in cardiac weight loss and preservation of anterior wall thickness	[50,51]
ROS Inhibitors	SS-31	Antioxidant that reduces ROS in mitochondria	Restored LV function, reduced proteolytic Calpain activity in heart	[52]
	Ubiquinol	Antioxidant involved in ROS modulation	Increased muscle mass; however, did not improve LV diameter or protein degradation	[53]
	Pepstatin	Inhibition of lysosomal protease and oxidative stress	Reduced muscle degradation, but no clear impact on myocardial function	[54,55]
Cardiovascular Drugs	Simvastatin	Decreases activity of matrix metalloproteinase-9 and reduces activity of various inflammatory markers	Decreased weight loss, improved LVEF, and increased SV	[56,57]
	Bisoprolol and spironolactone	Beta receptor blockade and aldosterone inhibitor respectively	Preserved LV mass, body weight; improved LVEF; reduced cardiac fibrosis	[55]
	Losartan and Withaferin A	Angiotensin II inhibition	Preserved EF and SV; reduced fibrotic deposition	[58,59]
	Formoterol	B2 selective agonist	Non-significant increase in cardiac weight; significant increase in end-diastolic and systolic volumes	[60]
Nutrition and Appetite	Rosiglitazone	Insulin sensitizer	Improved LVEF and cardiac output; decreased muscle wasting	[61]
	Megestrol acetate	Appetite stimulant	Increased weight gain; improved LVEF	[49]
	Leucine	Decreased levels of chymotrypsin, myeloperoxidase, and caspase 3 and 7	Improved myocardial function	[62]
	Lauric acid and Glucose	Reduced mitochondrial dysfunction and oxidative stress	Reduced myocardial atrophy and improved muscle maturity	[63]
	Total parenteral nutrition (TPN)	Intravenous nutrition supplementation	Increased cardiac mass	[64]
Other Categories	EPO	Possible decrease in trypsin levels	Increase in cardiac weight, stroke volume, and physical activity	[65]
	Oxypurinol	Xanthine oxidase inhibitor	Increased LVEF, total cardiac weight, and cardiac output	[66]
	Tandospirone	Antidepressant; serotonin receptor agonist	Preserved muscle mass, improvement in LV mass and EF; some survival benefit	[67]
	Testosterone	Unclear	Increase in SV and LVEF	[67]
	Exercise	Unclear; possible reduction in cardiac autophagy and NF-κB signaling	Impeded tumor growth, delayed onset of anorexia, improved EF	[68,69,70,71,72]
	Gene Therapy via Viral Vector	Upregulating SMAD7, known to inhibit overactivation of procachetic factors	Reduced skeletal and cardiac muscle atrophy	[73]
	Crytotanshinone	STAT3 inhibition	Decreased myocardial mass loss, body weight loss, and muscle wasting	[74]
	Minocycline	Matrix metalloproteinase inhibitor	Improved FS and EF	[75]

## Data Availability

Not applicable.

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
