# Peer review of "Review of Mechanisms and Treatment of Cancer-Induced Cardiac Cachexia"

_cells, 2022, doi:10.3390/cells11061040_

Round 1

Reviewer 1 Report

The manuscript “Review of Mechanisms and Treatment of Cancer-Induced Car-2 diac Cachexia“  by Vudatha et al. is a review that aims to summarize all the aspects regarding the cancer cachexia, a multifactorial and multi-organ syndrome characterized by sarcopenia and inflammation, typical in several chronic inflammatory disease that negatively impact patient quality of life. The authors focused their attention on cardiac cachexia associated to cancer.

I appreciated the review, that appears clear, well organized and well structured. Comparing more than 1000 papers, they have selected only 70 of them. The authors refer to recent papers and studies, from 1960 to 2021, so the review seems to be updated. The research of references is well described as well as the in vivo experiments and the clinical studies. The review deal with physiological and biochemical mechanisms associated to cardiac cachexia, animal models used, methods of diagnosis and disease treatments.

Few revisions are required:

  • Regarding the table 1, the treatments could be grouped according to drugs mechanism of action in order to make it as clear as possible for an immediate consultation.
  • I was wondering, why is pepstatin in table 1 if its benefits are unclear? Can you explain this aspect? Eventually, I suggest to shift pepstatin at the end of the table considering this uncertain aspect.
  • The references number 18 and 19 (regarding the role of TNF alpha), 30 (about the Ca2+ involvement), 45 (pepstatin paper), 71 and 74 (regarding pharmacological treatment and physical exercise), are not recent. The authors should update these specific parts of bibliography finding more recent studies.
  • The acronym LVEF is not extendedly explained. The authors have to write it extendedly the first time that they used the acronym.

Minor points:

Line 294, the bracket after inhibitor is missed,

Line 345, TNF-a has to be changed into TNF-α.

The manuscript is very interesting and useful; the publication is recommended.

Reviewer 2 Report

The article is generally well written and covers to an appropriate level of detail a variety of topics relative to cardiac cachexia in cancer patients. A few minor suggestions are listed below to help improve this work.

Page 3 (lines 123-130) is there information on the interaction of metformin or other biguanides and cachexia? Metformin use is associated with weight loss, in general…but there is a weak association with improved clinical outcomes. Please address this topic area.

Page 4 (lines 173-183) Activins have also been associated with EMT, which is associated with progressive disease and poor clinical outcomes. Is there any information relevant to Activins inducing EMT in the context of cachexia?

Page 4 (4.2. Models to study cardiac cachexia) Suggest adding descriptions of recent studies evaluating orthotopic tumors in murine models (i.e. PDX mice).

Page 4 (4.3. Methods to assess cardiac cachexia) Please discuss the potential role of skeletal muscle index and other measures integrating radiographic, clinical, and molecular components (CXI, NLR, etc.). Weight loss should also be addressed…relative to BMI. (advantages/ disadvantages)

Also, would like to see additional discussions on nutritional agents that may mechanistically service the other potential treatment options, including OTC agents (e.g., anti-inflammatory vitamin regimen)
